# Crystal structure of *E. coli* apolipoprotein N-acyl transferase

Guangyuan Lu[1,2,*], Yingzhi Xu[1,*,†], Kai Zhang[1,†], Yong Xiong[3], He Li[1], Lei Cui[1], Xianping Wang[1,2], Jizhong Lou[1,2], Yujia Zhai[1], Fei Sun[1,2] & Xuejun C. Zhang[1,2]

In Gram-negative bacteria, lipid modification of proteins is catalysed in a three-step pathway. Apolipoprotein N-acyl transferase (Lnt) catalyses the third step in this pathway, whereby it transfers an acyl chain from a phospholipid to the amine group of the N-terminal cysteine residue of the apolipoprotein. Here, we report the 2.6-Å crystal structure of *Escherichia coli* Lnt. This enzyme contains an exo-membrane nitrilase domain fused to a transmembrane (TM) domain. The TM domain of Lnt contains eight TM helices which form a membrane-embedded cavity with a lateral opening and a periplasmic exit. The nitrilase domain is located on the periplasmic side of the membrane, with its catalytic cavity connected to the periplasmic exit of the TM domain. An amphipathic lid loop from the nitrilase domain interacts with the periplasmic lipid leaflet, forming an interfacial entrance from the lipid bilayer to the catalytic centre for both the lipid donor and acceptor substrates.

[1] National Laboratory of Macromolecules, Institute of Biophysics, CAS Center for Excellence in Biomacromolecules, Chinese Academy of Sciences, 15 Datun Road, Beijing 100101, China. [2] School of Life Sciences, University of Chinese Academy of Sciences, 19 Yuquan Road, Beijing 100049, China. [3] Department of Molecular Biophysics and Biochemistry, Yale University, New Haven, Connecticut 06411, USA. * These authors contributed equally to this work. † Present addresses: Beijing Key Laboratory of Novel Technique for Combined Vaccine, Beijing Minhai Biotechnology Co. Ltd, 35 Simiao Road, Beijing 102600, China (Y.X.); Medical Research Council Laboratory of Molecular Biology, Division of Structural Studies, Francis Crick Avenue, Cambridge CB2 0QH, UK (K.Z.). Correspondence and requests for materials should be addressed to Y.Z. (email: yujia@ibp.ac.cn) or to F.S. (email: feisun@ibp.ac.cn) or to X.C.Z. (email: zhangc@ibp.ac.cn).

Lipoproteins are involved in membrane biogenesis as well as many other cellular functions in both prokaryotic and eukaryotic cells. Examples of well-known lipoproteins include components of LptDE, BamABCDE and LolBCDE complexes in biogenesis of the bacterial outer membrane and many potent activators of innate immune response[1–5]. In bacteria, lipoproteins are synthesized as pre-prolipoproteins in the cytosol. Each pre-prolipoprotein contains an N-terminal signal peptide which possesses a cysteine-containing 'lipobox' motif of a typical amino acid sequence of Leu–Ala–Gly–Cys. After being co-translationally translocated by Sec/Tat translocon systems to the periplasm, the lipidation of pre-prolipoprotein is sequentially catalysed by the following three enzymes to form a mature lipoprotein[6,7]: (i) diacylglyceryl (DAG) modification of pre-prolipoproteins by phosphatidylglycerol (PG):prolipoprotein DAG transferase (Lgt) to form prolipoproteins; (ii) cleavage of signal peptide from prolipoproteins by lipoprotein signal peptidase (LspA) to form apolipoproteins; and (iii) N-acylation of apolipoproteins by apolipoprotein N-acyl transferase (Lnt). While crystal structures of prolipoprotein DAG transferase and lipoprotein signal peptidase have been reported recently[8,9], the three-dimensional structure of Lnt remains unknown until now.

Lnt catalyses the transfer of the acyl lipid chain from the sn-1 position of a phosphoglycerol lipid-donor to the α-amine group of the N-terminal, DAG-modified cysteine residue of the apolipoprotein, completing the formation of a mature triacylated lipoprotein (Fig. 1a). On the basis of amino acid sequence analysis, Lnt belongs to the ninth of 13 branches of the nitrilase (Nit) superfamily[10,11]. Members of the nitrilase superfamily catalyse hydrolysis or condensation of carbon–nitrogen amine and nitrile bonds. The first reported crystal structures of soluble members of the nitrilase superfamily were that of N-carbamyl-D-amino acid amido-hydrolase (DCase) from Agrobacterium (protein data bank (PDB) ID: 1ERZ)[12] and a Nit-Fhit enzyme from Caenorhabditis elegans (1EMS)[13]. In addition, crystal structures of reaction intermediates have been reported for yeast Nit2 (yNit2, PDB IDs: 4HG3, 4HG5 and 4HGD)[14]. Characteristic structural features of this superfamily include a αββα type of folding topology and a set of conserved Glu–Lys–Cys catalytic triad (Fig. 2 and Supplementary Fig. 1)[15]. In Gram-negative bacteria, Lnt enzymes are integral membrane proteins located in the inner membrane, and are predicted to contain a Nit domain and a transmembrane (TM) domain consisting of at least six helices[11].

The Escherichia coli Lnt (Ec-Lnt, 57 kDa) consists of 512 amino acid residues, and based on sequence homology with other Nit proteins its catalytic triad is identified as E267-K335-C387 (ref. 11). Lnt requires two substrates, namely an acyl donor and an acceptor (that is, apolipoprotein). Biochemical analysis showed that Ec-Lnt is able to utilize all phospholipids present in E. coli, including negatively charged PG and cardiolipin, and electro-neutral phosphatidylethanolamine, as acyl donors[16]. Using an in vitro assay, it was shown that Lnt uses a ping-pong mechanism, including slow formation of acyl-enzyme intermediate (that is, the first step) and rapid N-acyl transferring to the apolipoprotein (the second step)[17]. Based on a current mechanistic model of the nitrilase superfamily[12], E267 of Ec-Lnt serves as a general base to activate nucleophile (that is, the thiol group of C387) in the first step and to activate amine group in the second step of the transacylation reaction; C387 forms a thiol-acyl intermediate; and K335 provides part of the oxyanion hole to stabilize the tetrahedral intermediate of the reaction carbon[18].

Here we set out to determine the structure of E. coli Lnt using X-ray crystallography to address the question of how the Nit and TM domains are arranged within the Lnt structure, and to determine how substrates access the catalytic site of Lnt. Our analysis shows that Lnt contains an 8-helix TM domain, with the

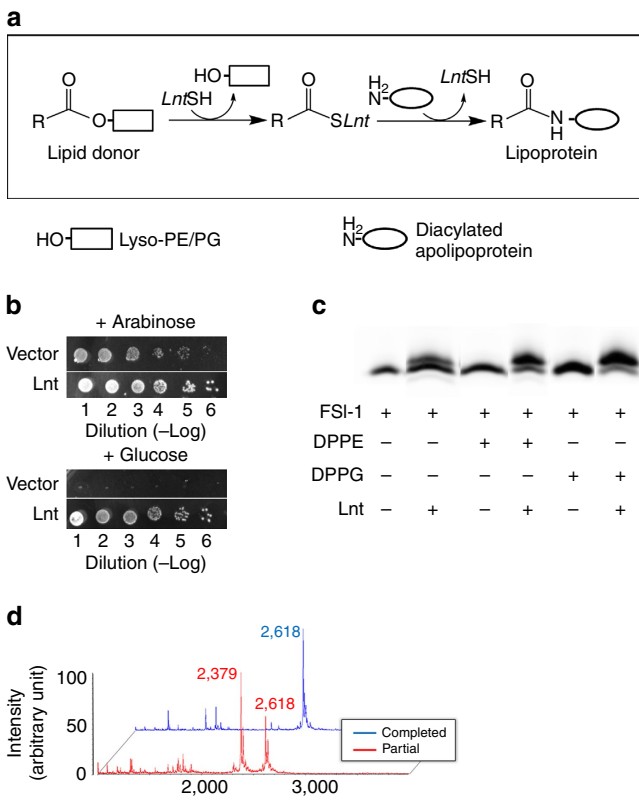

**Figure 1 | Functional assays to assess recombinant Ec-Lnt functions.**
(**a**) Schematic diagram of the two-step reaction catalysed by Lnt (ref. 10). (**b**) Complementation assay of Lnt with the Δlnt cell strain. The cell strain contains two plasmids. One serves as a conditional rescuing plasmid and carries the WT Lnt gene which is induced by arabinose but inhibited by glucose. The other plasmid contains constitutively expressed Lnt gene to be tested. A serial dilution of cell culture was spotted onto solid medium containing either 11 mM glucose or 1 mM arabinose. Images were taken 16 h after incubation at 37 °C. 'Vector' stands for a plasmid without the Lnt gene. (**c**) In vitro transacylation assay. Lnt (3.5 nM) was added to a reaction mixture composed of lipids (2.5 μM) and FSL-1 (40 μM). Samples were incubated for 40 min at 37 °C and analysed by Tricine-SDS–polyacrylamide gel electrophoresis. The lower and upper bands are unmodified and modified substrates of FSL-1 fluorescein, respectively. (**d**) MALDI-TOF mass spectroscopy analysis of the in vitro reaction product. During the full reaction (that is, with both the lipid donor and acceptor), the peak of substrate FSL-1 fluorescein moved from 2,379 Da (red peak) to 2,618 Da (blue peak), with their mass-difference matching well with the calculated value of 239 Da for the acyl chain of palmitic acid.

Nit domain located between TMs 7 and 8. The TM domain provides a lateral opening for the substrates, and the catalytic Nit domain caps the TM domain from the periplasmic side.

## Results

**Cavities of the TM and Nit domains are connected.** Ec-Lnt was over-expressed in E. coli as a His$_8$-tagged recombinant protein. Complementation of lnt-conditional depletion cells[15] confirmed that our recombinant Ec-Lnt possessed in vivo activity (Fig. 1b). To investigate the in vitro activities of recombinant Ec-Lnt, we performed mass spectroscopy and gel-shift analysis of the full reaction (Fig. 1c,d). Our results clearly demonstrate that the purified Ec-Lnt possesses acyl transferase activity (see below).

Ec-Lnt was purified and crystallized in the presence of 0.3% (w/v) n-nonyl-β-D-thiomaltopyranoside (NTM). The crystal structure

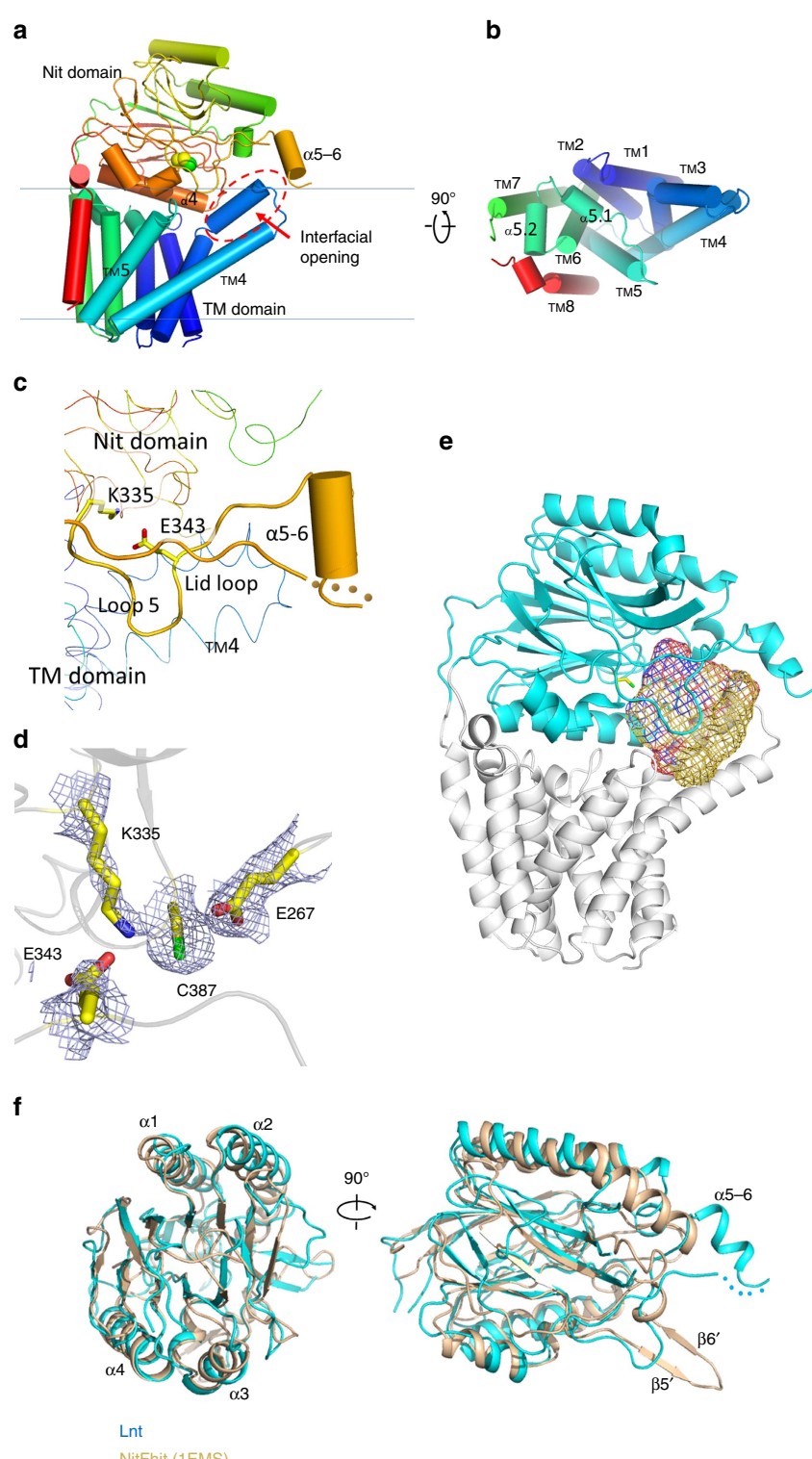

**Figure 2 | Structure of *E. coli* Lnt.** (**a**) The peptide chain of Lnt is shown in rainbow scheme from the N- (blue) to the C-terminus (red). The sidechain of the catalytic Cys387 is shown as sphere model, helices are shown as cylinders and β-strands are shown as arrows. The putative substrate entrance of the interfacial opening is marked with the red oval. (**b**) The TM domain is viewed from the periplasmic side, with the Nit domain removed for clarity. (**c**) Cartoon presentation of the lid loop and surround region. The missing residues ($F^{358}DL^{360}$) in the lid loop are represented by dots. (**d**) Catalytic centre. Sidechains of the catalytic triad and Glu343 are shown in stick model. The 2Fo-Fc electron density map is contoured at 1-σ level. (**e**) Substrate-binding cavity. The Nit and TM domains are shown as cyan and grey ribbons, respectively. The cavity was calculated with the programme EdPDB and shown as mesh using the programme PyMol. Cavity surface regions in contact with oxygen and nitrogen atoms are coloured in red and blue, respectively; otherwise in yellow. The catalytic residue Cys387 is shown as a stick model. (**f**) The two Nit domains are superimposed, and shown in ribbon presentations. *E. coli* Lnt is coloured in cyan, and Nit-Fhit (PDB ID: 1EMS) is in wheat colour. The root mean square deviation (r.m.s.d.) between the Nit domains of Ec-Lnt and *C. elegans* Nit-Fhit is 1.5 Å (for 175 Cα atom pairs).

## Table 1 | Statistics of data collection and refinement.

| Crystal | Lnt-native | Lnt (Se-peak) |
|---|---|---|
| *Data processing* | | |
| Beamline | BL17U at SSRF | BL1A at KEK |
| Wavelength (Å) | 0.9793 | 0.9791 |
| Space group | $P2_12_12_1$ | $P2_12_12_1$ |
| *Cell dimensions* | | |
| a, b, c (Å) | 91, 134.1, 135.2 | 92.5, 134.1, 134.5 |
| $\alpha, \beta, \gamma$ (°) | 90, 90, 90 | 90, 90, 90 |
| Resolution (Å)* | 50 − 2.59 | 50 − 3.60 |
| | (2.64 − 2.59) | (3.69 − 3.60) |
| Completeness (%) | 100 (100) | 84.4 (74.3) |
| $R_{merge}$ (%) | 13.5 (66.5) | 16.3 (42.7) |
| $I/\sigma(I)$ | 14.8 (2.5) | 8.3 (2.3) |
| Unique reflections | 51,623 (1,457) | 23,977 (1,084) |
| Redundancy | 7.2 (6.6) | 5.7 (5.3) |
| | | |
| *Refinement* | | |
| Resolution (Å) | 40.4-2.59 | |
| Number of reflections (test)† | 42,099 (2,138) | |
| $R_{work}/R_{free}$ (%) | 23.2/28.9 | |
| Average B-factor (Å²) | 46.8 | |
| *R.m.s.d. from ideal geometry* | | |
| Bond lengths (Å) | 0.010 | |
| Bond angles (°) | 1.445 | |
| *Ramachandran plot (%)‡* | | |
| Favoured | 94.1 | |
| Allowed | 5.9 | |

*Values in parentheses are for the highest resolution shell.
†Reflections of the test set were selected before refinement.
‡Calculated using *MolProbity*.

of apo-Ec-Lnt was determined using a combination of molecular replacement (MR) and single anomalous dispersion (SAD) methods, and was refined at 2.6-Å resolution. N-terminal residues $1 − 8$, C-terminal residues $505 − 512$, and two other segments of residues $305 − 309$ and $358 − 360$ were missing from the refined model because of lack of interpretable electron densities. Statistics of data collection of the X-ray diffractions and of refinement of the crystal structure are summarized in Table 1.

Unlike most reported soluble nitrilase proteins forming homo-dimers or oligomers, Ec-Lnt exists in a monomer form in our crystal structure as well as in detergent solubilized form in solution (Supplementary Fig. 2). Ec-Lnt contains a 7-TM N-terminal domain, a Nit domain (that is, residues $T215 − T478$), and a C-terminal helix TM8 (Fig. 2 and Supplementary Fig. 3). Together, TMs $1 − 7$ and TM8 form the TM domain, with both the N- and C-termini located on the cytosolic side. The TM-Nit domain interface buries a total surface area of 3,700 Å². The interface on the TM-domain side is concave and extensive, including the two short helices α5.1 and α5.2 between TM5 and TM6. On the Nit-domain side of the domain interface, α-helices α3 and α4 are involved in the domain packing. A likely conserved hydrogen bond, Y153–R432, is formed between α5.1 of the TM domain and α4 of the Nit domain (Fig. 3). Mutation of Y153A showed reduced activity. Furthermore, the Nit domain is partially docked into the TM domain, bringing the catalytic C387 residue close to the lipid bilayer. In agreement with this arrangement, TMs 5 and 6 are shorter than other TM helices. In contrast, TMs 3 and 4 are longer, promoting a tilting conformation of the TM domain relative to the membrane. TM4 is estimated to be located 45° off the membrane normal. The TMs 3, 4, and 5 form a membrane embedded, hydrophobic cavity (Fig. 2b and Supplementary Fig. 4), with (i) its upper opening facing the active-site pocket of the Nit domain and (ii) a lateral opening to the membrane bilayer.

**The Nit domain of Lnt is structurally similar to Nitrilase.** Similar to other members of the nitrilase superfamily, the Nit domain of Lnt contains two major six-stranded β-sheets, forming the core of the Nit domain. Following the terminology used for the Nit domain of *C. elegans* Nit-Fhit (PDB ID: 1EMS)[13], the two β-sheets are composed of β12-β1-β2-β3-β4-β5 and β6-β7-β8-β9-β10-β11 (Supplementary Fig. 1b). The Nit domain of Lnt superimposes well with that of known nitrilases (Fig. 2f). However, significant shortening occurs in Lnt, in two regions between β2 and α2, and between β9 and β10 (Supplementary Fig. 1b). Both of these truncations make room for the following additional changes: Secondary structure elements between β5 and β6 in Nit-Fhit (including a β-hairpin formed by β5′ and β6′) are replaced in Lnt by a ∼35-residue moiety (that is, residues $N336 − Y370$), including a short helix (termed α5-6), a preceding loop-5, and a following, flexible, amphipathic loop (termed lid loop; Fig. 2c). The lid loop extends from the Nit domain and reaches the tips of the loop connection TMs 3 and 4 of the TM domain. Part of the lid loop (that is, F358-D359-L260) is among the missing parts in the refined structure model. We speculate that α5-6, together with the lid loop (including the sidechain of the missing F358), anchors the Nit domain to the periplasmic leaflet of the inner membrane under *in vivo* conditions. Consequently, the main body of the Nit domain, the loop-5, α5-6, the lid loop and TMs 3–5 from the TM domain form a dome-shaped cavity (∼2,000 Å³) in the vicinity of the catalytic C387 (Fig. 2e). Residues involved in forming this cavity include catalytic triad and regions of A72–L101, L143–W148, N336–S364, Y388 and D413–F416 (marked with green dots in Supplementary Fig. 3). The cavity maintains an opening that faces the lipid bilayer. This interfacial opening is likely to serve as the entry for both the donor- and acceptor-substrates to gain access to the catalytic site from the lipid bilayer.

The catalytic site of Ec-Lnt is located in a deep pocket inside the Nit domain. The conformation of the entire catalytic triad, E267-K335-C387, is nearly identical to that of soluble members of the nitrilase superfamily (Supplementary Fig. 1c). The general base E267 is stabilized by a highly conserved N314 by way of a hydrogen bond formed between the two residues. C387 is located at the N-terminal of a short, distorted helix (residues C387–L392), termed α3′. The N-terminal region of α3′ contributes to the formation of the oxyanion hole, while the C-terminal end is capped by a conserved basic residue, R432. A conserved acidic residue in α3′, E389, forms a hydrogen bond with Y333, stabilizing the position of α3′. Moreover, catalyst K335 is stabilized by the so-called fourth catalytic residue[19], E343, located in the loop-5 (Fig. 2d).

**Essential Lnt residues are located within the central cavity.** To screen functionally important residues in Ec-Lnt, a complementation assay was used. While wild-type (WT) Ec-Lnt readily rescues the conditional depletion strain, termed Δ*lnt* (Fig. 1b; see the Methods section), variants containing a disruptive mutation at important positions showed no such effects. Over 40 positions located inside the central cavity were subjected to Ala-mutation and screened for their ability to rescue Δ*lnt* (Fig. 3 and Supplementary Fig. 5a). As expected, mutations at the catalytic triad and E343 lost activity. In addition, we found several other positions whose Ala-mutations lacked the ability to rescue Δ*lnt* cells, namely G145, V339, G342, R352, Y388, E389 and F416. Their expression levels were verified to be comparable to those of WT (Supplementary Fig. 5b). Among them, G145 is located in a highly conserved region C-terminal to TM5. All the remaining positions identified belong to the Nit domain. Both V339 and G342 are located in the loop-5, and their mutation may negatively influence the catalytically important E343. The sidechain of R352

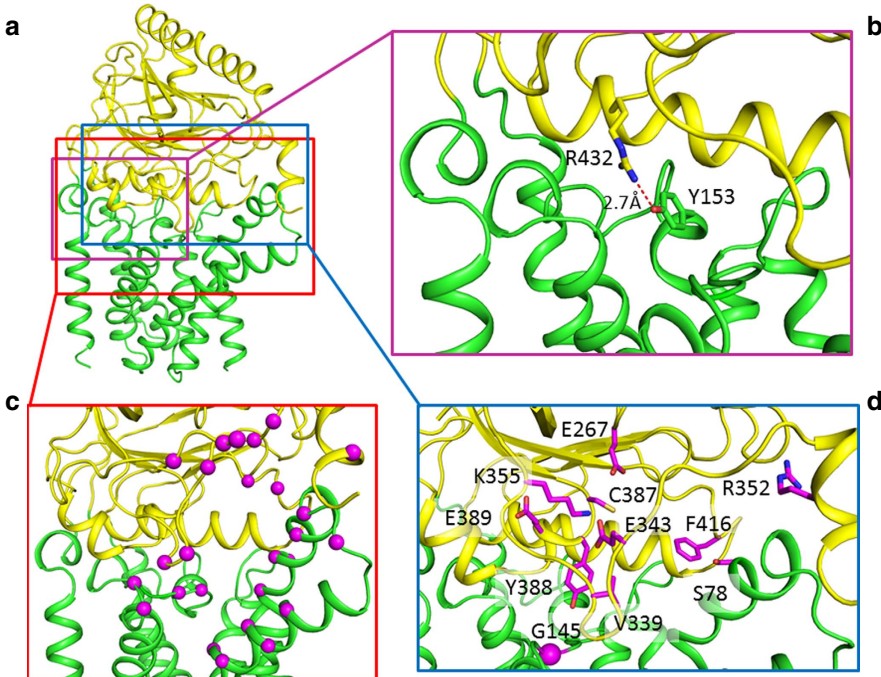

**Figure 3 | Distribution of mutations sites in the three-dimensional structure of Lnt.** (**a**) Overall structure of Lnt. The peptide backbone is shown as a tube model, with the TM and Nit domains being shown in green and yellow, respectively. (**b**) The interface between the Nit and TM domains. (**c**) Positions where a single Ala-mutation allows Lnt to maintain its activity are shown as magenta spheres. (**d**) Sidechains of residues for which a single Ala-mutation renders Lnt inactive are shown in stick models (Gly as a sphere) and coloured in magenta.

from the Nit domain has multiple contacts with surrounding residues, and thus the activity loss of R352A mutation is more likely to be related to structural disruption. Y388 is located next to the catalytic residue C387, and mutation Y388A may disrupt both substrate binding and the conformation of C387. As mentioned above, E389 (conserved in Lnt homologues) forms a hydrogen bond with Y333, stabilizing α3′. F416 is located in a conserved loop C-terminal to β8 of the Nit domain. To further identify functionally important residues, we constructed additional tryptophan mutations at selected positions inside the putative substrate-binding cavity (Supplementary Fig. 5a). As a selection criterion, the tryptophan sidechain to be introduced must fit inside the cavity without disrupting in the surrounding crystal structure. Among these tryptophan mutations, S78W and Y388W were found to be inactive. In particular, S78 is located at the C-terminus of TM3 and is near the interface between the Nit domain and the TM domain.

Since the transacylation reaction uses a ping-pong mechanism, an inactive mutation may specifically disrupt either the first or second step of the acyl-transferase reaction. To determine functional roles of the residues that are associated with inactive variants, we performed mal-PEG (that is, maleimide-conjugated PEG 5000) modification analysis on the above identified inactive variants. Mal-PEG may react with free thiol groups, thus determining whether a mutation is able to form the acyl-enzyme intermediates[18]. For this purpose, we first constructed a C23A/C62A mutant (termed as WT*), which eliminated free thiol groups except that of the catalytic C387. This WT* variant was shown to maintain the WT activity (Supplementary Fig. 5a) and was used subsequently as the background construct in the mal-PEG assay. Among the inactive mutation variants tested, S78W (in TM3), V339A (lid loop), Y388W (nearby the catalytic Cys residue) and E389A (α3′) were found to maintain the ability to form the acyl-enzyme intermediate (Fig. 4 and Supplementary Fig. 7), suggesting that the first step of the reaction was not disrupted by these mutations. Thus, the corresponding residues in

the WT Lnt are more likely to be involved in the second step of the reaction, probably in recognizing and binding of the prolipoprotein substrate.

**Lnt activity depends on its affinity to the lipid substrate.** To analyse the specificity of lipid-donor substrates, we performed both thermofluor assays[20] and *in vitro* gel-shift assays using S-[2,3-bis(palmitoyloxy)-(2RS)-propyl]-(R)-cysteinyl-GDPKHPKSFK (fluoresceinyl-ε-aminocaproyl-ε-aminocaproic acid) (abbreviated as FSL-1) as the lipid acceptor[17]. In the thermofluor assay, PG molecules showed the most significant effects on stabilizing Lnt (Supplementary Fig. 8a and Supplementary Table 1). Furthermore, PG molecules also appeared to be good substrates for Lnt (Supplementary Fig. 8b). Together, these findings provide evidence of a strong correlation between the substrate affinity and the ability for the lipid molecule to serve as an acyl-chain donor.

**The lid loop interacts with the lipid bilayer.** To assess the functional role of the lid loop, we carried out molecular dynamic (MD) simulations. A total of three simulations were performed, with distinct initial positioning of the lid loop relative to the membrane. In one simulation, the lid loop was buried inside the outer leaflet of the membrane (simulation 1, or Sim 1). In comparison, in simulations Sim 2 and 3, the loop was initially placed outside the membrane and exposed to the solvent phase. Each system was simulated for 150 ns. In all cases, the overall structure of Lnt was maintained, with the backbone root mean square deviation (RMSD) below 2.5 Å throughout the duration of all simulations (Supplementary Fig. 6a). The protein maintained a tilted orientation relative to the membrane plane, with the cytosolic portion of the curved TM4 being located about 45° off the direction of the membrane normal, and with the periplasmic portion of TM4 being 70° off. Not surprisingly, our simulations indicated that the most flexible region of the entire Lnt protein is the lid loop. After aligning the remaining part of the Nit domain with the initial structure, the RMSD change of the lid loop in

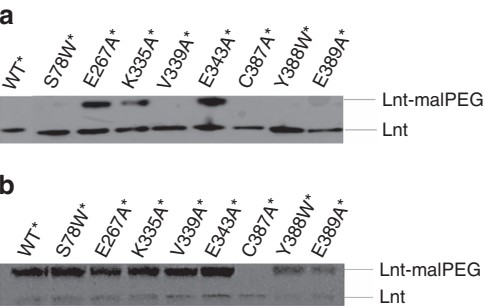

**Figure 4 | Effects of different mutated residues on the acylation activity of Lnt.** Eight key residues that caused dysfunction of Lnt in the *in vivo* assay were combined in the C23A/C62A variant of Lnt (that is, WT*) and were analysed by mal-PEG alkylation. Lnt (C23A/C62A/C387A), that is, C387A*, lacks any cysteine residue and was used as a negative control. Whole-cell lysates were treated without (**a**) or with (**b**) 1 M neutral hydroxylamine (which reduces all thioester bonds) and subsequently treated with mal-PEG. Samples were analysed by SDS-10% polyacrylamide gel electrophoresis followed by immunoblotting with antibodies against polyHis-tag. Alkylated (Lnt-mal-PEG) and non-alkylated forms of Lnt are indicated. Note that, variants E267A, K335A, E343A, C387A and E389A have been reported to be inactive using the same assay[18].

Sim 1, converged to ~6 Å (Supplementary Fig. 6b). Structure comparison between the last snapshot of the simulations and the initial structure is shown in Supplementary Fig. 6c, and the flexibility of the lid loop is significantly larger than the remaining structure of the enzyme (Supplementary Fig. 6d). For Sim 2 and Sim 3, the RMSD of the lid loop converged to ~8 Å, the lid loop remained outside of the membrane through the simulations, and the α5-6 became less regular.

The results of MD simulation were in agreement with the suggested possibility that Nit domain is partially embedded into the lipid membrane, especially the lid loop (Fig. 5). In Sim 1, the short helix α5-6 (E348–A355) immerged into the bilayer. This suggests that the presence of a lipid bilayer stabilizes the α5-6 helix. Importantly, the lid loop anchored the Nit domain to the outer leaflet of the lipid bilayer through multiple interactions, including those between lipid molecules and the sidechains of F357 and F358; between the acidic sidechain of D359 and the head-group of a PG molecule; and between the basic sidechain of R352 and the phosphate-group of a lipid molecule (Fig. 5b). The aromatic ring of F357 is ~10 Å below the level of the phosphate groups of the lipid molecules. In agreement with the MD stimulation, our complementation assay showed that the double mutant form F357A/F358A is inactive (Supplementary Fig. 5a). In addition, our MD simulation results showed that in the vicinity of lipid bilayer-protein interface, including the α5-6 region, the membrane surface adopts a concave shape (Fig. 5a). Interestingly, in our Sim 1 simulation, a lipid molecule enters the cavity during the simulation (Supplementary Movie 1), although its head-group remained ~13 Å always from the catalytic C387 at the end of the simulation. This observation supports the notion that the cavity is able to accommodate lipid substrate (Fig. 5a).

## Discussion

Our crystallographic analysis of Ec-Lnt reveals an overall structure of an 8-helix TM domain capped by the Nit catalytic domain from the periplasmic side. We found that P147, a previously identified important residue for Lnt activity[21], is located at the interface between the TM and Nit domains. This suggests that to ensure full Lnt activity, proper docking of the Nit domain to the TM domain is essential. Such geometric complementation between the two domains also explains why the

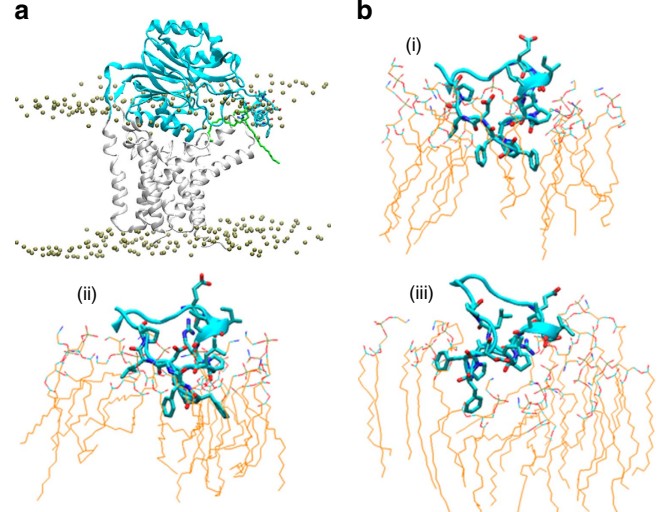

**Figure 5 | Snapshot of the last step of simulation 1.** In this simulation (Sim 1), the lid loop is embedded into the lipid membrane. (**a**) Overall conformations of Lnt and surrounding lipid molecules. The Nit and TM domains are shown as cyan and grey cartoon model, respectively, and the lid loop region is shown in cyan stick model. For the lipid bilayer, only the phospho-atoms from the lipid molecules are shown as spheres, except for one lipid molecule inside the cavity, which is shown as green stick model. (**b**) More detailed interactions between lid loop residues (thick stick models) and surrounding lipid molecules (thin line models) for three snapshots. (i) The initial energy-minimized model; (ii) a snapshot after 20 ns simulations with Lnt constrained; (iii) a snapshot after 150-ns free dynamics simulations. (Also see a Supplementary Movie 1).

periplasmic loop between TMs 5 and 6 is not accessible to chemical modification from the periplasmic side[21].

The catalytic triad is highly conserved between Lnt proteins and those of soluble Nit proteins. Based on structural comparison, we speculate that the thioester Lnt-acyl intermediate has a local structure nearby the catalytic C387 similar to that of the yNit2 (enzyme)-αKG (substrate) complex, with the lipid chain protruding into the lipid bilayer via the interfacial opening (Supplementary Fig. 7). Furthermore, the active-site cavity of the Nit domain is formed by conservative residues of the Lnt family (Supplementary Fig. 3). C-terminal to β5 (and the catalytic K335), the loop-5 (Asn336–Leu347) region is also highly conserved. This loop contains the fourth catalytic residue E343 and forms part of the rim of the interfacial opening. Our E343A mutation confirmed its importance in the first step of reaction, that is, formation of the enzyme-lipid intermediate. While this acidic residue is not conserved within the nitrilase superfamily, it is highly conserved in Lnt proteins, suggesting that it may be required specifically for Lnt-associated functions (for example, lipid recognition). In addition, the Nit-TM domain interface, including α5.1 and α5.2 of the TM domain and α4 from the Nit domain, is also among the conserved regions. Therefore, we predict that the observed overall structure of Ec-Lnt is conserved in other Lnt proteins.

Of particular interest is the flexible, amphipathic, lid loop (residues F357–Q372) located at the C-terminal of the short helix α5-6. This region exhibits significant variation in its primary sequence compared with other proteins of the Nit family (Supplementary Fig. 1). Part of this region is invisible in the crystal structure, presumably because of high flexibility in the absence of lipid bilayer. Our results showing that the F357A/F358A mutation is inactive are in agreement with previous results for the importance of the lid loop region[21].

Although the sequence of this region is not strictly conserved among Lnt proteins, its physicochemical nature remains largely amphipathic. Thus, we propose that this lid loop interacts with the lipid membrane where it forms a lid protecting the catalytic site from exposure to the solvent phase. Such hypothesis is supported by our MD simulation data, which provide strong evidence that the lid loop has a favourable interaction with the lipid bilayer. Because of this lid loop, the catalytic cavity of the Nit domain can only be accessed by the substrates from a membrane-embedded cavity formed by TMs 3, 4 and 5. This later cavity has a lateral opening to the lipid bilayer. Similar lateral openings are common amongst membrane integral enzymes[8,9,22,23], presumably because they are crucial for proper access of lipophilic substrates and/or products to the active site.

Lnt catalyses the transacylation from the sn-1 position of the acyl-donor PG to the α-amine group of the N-terminal Cys residue of the substrate peptide via a two-step, ping-pong reaction[18]. Our new structure information supports such a ping-pong mechanism (Supplementary Fig. 9): the lipid donor–substrate enters the catalytic cavity of the Nit domain through the interfacial opening. After formation of the thioester acyl-enzyme intermediate, the leaving group sn2-lyso-glycerolphosphate is released to the lipid bilayer. The lipid-acceptor substrate then enters the reaction cavity and uses its N-terminal amine group to attack the reaction-centre carbon atom, resulting in cleavage of the C–S bond of the reaction intermediate. In conclusion, our 2.6 Å crystal structure of Ec-Lnt provides a solid basis for future structure-function studies aimed at identifying the molecular details of Lnt substrate recognition.

## Methods

**Construction of Lnt variants and plasmids.** The full-length *lnt* gene (GenBank ID: 946201) was amplified from the *E. coli* K-12 genome using PCR. Lnt variants were individually amplified and ligated into the pET-22b(+) based vector. Each recombinant plasmid encodes for an Lnt variant fused with a C-terminal His$_8$-tagged for purification and immunoblotting. Plasmids of all variants were sequenced through the entire Lnt gene.

**In vivo complementation activity assay.** In vivo activity assay was performed using $P_{ara}$-*lnt*$^{Ec}$ *E. coli* strain PAP8504 (ref. 15) (denoted as Δ*lnt*), which is a conditional *lnt* variant with an arabinose promoter, $P_{ara}$, inserted upstream of the chromosomal *lnt* gene. Addition of arabinose to the medium allowed growth of the Δ*lnt* strain, whereas addition of either glucose or fucose repressed expression of the Lnt and resulted in cell lysis. pEXS-DH plasmids carrying Lnt variants were transferred individually into Δ*lnt* cells. Transformed cells were selected on an LB agar plate containing arabinose, kanamycins and ampicillin. To test for complementation ability of Lnt mutants, single colonies were re-streaked onto plates containing 0.2% (w/v) arabinose or 0.4% (w/v) glucose.

**Expression and purification of recombinant Lnt.** Each recombinant protein was expressed in the *E. coli* strain C41 (DE3) in 2 × YT medium. IPTG (0.5 mM final) was added and then cultured for 16 h. Cells were centrifuged, and pellets lysed by using high pressure homogenization in lysis buffer (50 mM HEPES (pH 7.5), 100 mM NaCl, 10% (v/v) glycerol, 20 mM imidazole and 1 mM PMSF) and centrifuged at 18,000g for 20 min to remove debris. The membrane fraction was collected by centrifuging the supernatant at 100,000g for 1 h. The membrane pellet was then dissolved in lysis buffer containing 2% (w/v) n-decyl-β-D-maltopyranoside at 4 °C for 1 h. The insoluble fraction was removed by centrifugation at 100,000g for 30 min. Tagged proteins in the supernatant were purified by Ni-affinity chromatography followed by desalting and ion-exchange chromatography in HEPES (25 mM, pH 7.5) buffer containing 0.2% (w/v) n-decyl-β-D-maltopyranoside. Lnt fractions were eluted from the ion-exchange column at ~300 mM NaCl, which were then pooled and concentrated to 0.5 ml by ultrafiltration with molecular-weight cutoff 50 kDa (Millipore), before being loaded onto a Superdex 200 column (10/300 GL; GE Healthcare) pre-equilibrated with a mobile phase containing 25 mM MES (2-(N-morpholino)ethanesulfonic acid; pH 5.5), 100 mM NaCl, 1 mM EDTA, 1 mM DTT and 0.3% (w/v) NTM. The column effluent was pooled and concentrated to ~12 mg ml$^{-1}$ for crystallization trials or stored at −80 °C.

**Crystallization.** Crystallization of Lnt was performed at 16 °C using the hanging-drop vapor-diffusion method. Hexagon shaped crystals with a maximum size of

300 × 100 × 20 μm$^3$ grew in 50 mM Tris-HCl (pH 7.5) and 26% (v/v) PEG 550 MME supplemented with two detergents (n-Heptyl-β-D-thioglucopyranoside and CHAPSO). Seleno-methionine (Se-Met) Lnt was expressed in *E. coli* methionine-auxotrophy strain B834 (DE3) and purified with the same procedures as native Lnt except that 5 mM DTT was present during the entire purification process.

**Data collection and structure determination.** Crystals of Lnt were soaked in a cryoprotectant consisting of 50 mM Tris-HCl (pH 7.5), 30% PEG 550 MME, 0.15% (w/v) NTM and 0.85% (w/v) n-Heptyl-β-D-thioglucopyranoside, and were flash-cooled at 100°K in a stream of nitrogen gas. Diffraction data of Lnt crystals were collected at 100°K on beamline BL17U, Shanghai Synchrotron Radiation Facility (SSRF, China). SAD diffraction data of Se-Met derivative were tested on beamline BL41XU, Spring 8 (Japan) and collected at 100°K on beamlines BL1A and BL17A, Photon Factory (KEK, Japan). Data were processed using *HKL*-2000 package[24].

Positions of Se atoms were determined using SHLEXD[25]. Initial SAD phases from the Se-Met data set were calculated using the programme SOLVE and refined with RESOLVE[26]. A model of the Nit domain was built using Phyre2 and used as a partial search model for MR. The non-crystallography symmetry (NCS) averaging and anisotropy correction[27] (http://services.mbi.ucla.edu/anisoscale/) were used for further improvement of the map for initial model building. The model that best fitted the experimental density was combined with the manually placed TM helices for better phasing. The MR phases were updated iteratively with model building. To avoid model-bias, MR phases were combined with experimental phases in each cycle of model building. The final model was refined at 2.6-Å resolution using the programme Phenix[28].

**In vitro transacylation activity assay.** To measure the in vitro transacylation activity of Lnt variants, an Lnt activity assay was conducted using the lipo-peptide FSL-1 (EMC microcollections, Germany) as the lipid-acceptor substrate. FSL-1 exhibits an excitation maximum of 494 nm, and an emission maximum 517 nm. Reactions were set up in a HEPES buffer (50 mM, pH 7.5) containing 5 mM DPPG (di-palmitoyl-phosphatidyl-glycerol; or other specified lipids), 5 μM FSL-1 as well as 0.2% (w/v) DM. After a single short sonication step at 37 °C, 6 μM (final; ~0.5 μl from 7 μg μl$^{-1}$ store) of Lnt was added to the reaction mixture (with a final volume of 10 μl) and incubated at 37 °C for 40 min. Then FSL-1 samples were analysed with high-resolution Tricine-buffered SDS–polyacrylamide gel electrophoresis[29], and images taken using a Typhoon FLA 7000 scanner (GE Healthcare).

**Alkylation assay using mal-PEG.** The presence of free thiol groups in WT Lnt, the single cysteine Lnt (that is, the C23A/C62A mutant), and other Lnt variants was analysed by measuring their alkylation levels using maleimide-conjugated polyethylene glycol 5000 (Mal-PEG) as described previously[18]. Following induction with 1 mM IPTG at 16 °C overnight, 1 ml of bacterial culture was centrifuged at 16,000g, and denatured for 30 min by addition of 0.5% SDS in 1 M Tris-HCl (pH 8.0), either with or without 1 M hydroxylamine. Mal-PEG was added to a final concentration of 2 mM and incubated for 30 min at 25 °C. Since Mal-PEG added ~5-kDa molecular weight to Lnt, alkylated proteins were identified by SDS–polyacrylamide gel electrophoresis followed by immunoblotting.

**Thermostability assay.** To evaluate the effects of lipid binding on the stability of Ec-Lnt, thermofluor analysis was performed using a quantitative PCR instrument (Rotor-Gene 6600, Corbett Research, Australia). As temperature rose, thiol-specific fluorochrome N-[4-7- (diethylamino-4-methyl-3-coumarinyl)phenyl]maleimide (CPM; from Invitrogen, US) was used as fluorescence probe to monitor the thermal denaturation of Lnt. CPM fluorescence exhibits an excitation maximum at 387 nm and an emission maximum at 463 nm. Purified Lnt protein (10 μg μl$^{-1}$) was added into the reaction buffer (100 mM MES (pH 6.0), 100 mM NaCl and 25 μM CPM) in the presence or absence of 100 μM lipids to be analysed. Reaction volume was 25 μl, and the final concentration of Lnt was 0.4 mg ml$^{-1}$. The quantitative PCR instrument was programmed to increase temperature by 1 °C per min, and then to stay at each temperature for 1.5 min between 25 °C and 95 °C. Data analysis was performed using the programme Graphpad Prism.

**Molecular dynamics simulations.** A solvated Lnt structure was used as the starting point for MD simulations. The missing loops (including part of the lid loop region) in the structure were added with ModLoop[30]. The resulting structure was then inserted in a pre-equilibrated lipid bilayer containing DPPE (di-palmitoyl-phosphatidyl-ethanolamine) and DPPG (at a ratio of 4:1 which is initially constructed with CHARMM-GUI membrane builder (www.charm-gui.org)) to mimic bacteria membrane. Hydrogen atoms were added to the model system, and the system was solvated with a water box sized 100 × 100 × 120 Å$^3$. Counter-ions Na$^+$ and Cl$^-$ were added to neutralize the system and to maintain a salt concentration of 150 mM. The crystal structure of Lnt protein was first inserted into the membrane following procedures described in NAMD membrane protein tutorial (http://www.ks.uiuc.edu/Training/Tutorials/science/membrane/mem-tutorial.pdf), which generated an initial configuration (Sim2). Then the model was rotated 5° about an axis (perpendicular to membrane) manually in two opposite directions to move the lid loop into or away from the membrane surface, resulting

in initial configurations for Sim 1 and Sim 3, respectively. By using different membrane patches, three model systems were built independently, with slightly different protein orientations relatively to the membrane. In total, each of the initial model systems contained ~110,000 atoms.

The MD simulations were carried out with the NAMD 2.10 package[31] using CHARMM27 force field with CMAP corrections for proteins[32], CHARMM36 force field for lipids[33] and TIP3P for water molecules. Initially, the energy of the systems was minimized step by step with (i) all heavy atoms fixed, (ii) heavy atoms of the protein and lipids fixed, (iii) backbone atoms of the protein fixed and then (iv) the Cα atoms of the protein fixed. Subsequently, a 20-ns simulation was carried out, with the Cα atoms of the protein fixed. During this 20-ns simulation, water molecules were monitored using a script to prevent these molecules from penetrating the lipid bilayer. A 10,000-step energy minimization was then carried out with all constrains removed. The system was then simulated for ~150 ns for each of the three models. In all simulations, periodic condition was enforced, a cutoff of 12 Å was used in the calculation of non-bounded interactions, and a Partical–Mesh–Ewald method was used for calculating the electrical static interactions. The temperature of the system was constrained at 310 °K with Langevin dynamics, and the pressure of the system was maintained at 1 atm with Nosé-Hoover-Langevin piston pressure control. A rigid bond algorithm was used to allow simulation step size of 2 fs. The simulated trajectories were analysed with the programme VMD (ref. 34).

**Data availability.** Coordinates of the crystal structure of Lnt were deposited into PDB. The accession code is 5XHQ. The data that support the findings of this study are available from the corresponding authors upon reasonable request.

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

## Acknowledgements

We would like to thank the staff of the Protein Research Core Facility at the Institute of Biophysics, Chinese Academy of Sciences (CAS) for their excellent technical assistance. We are grateful to staff members of SSRF (China) Spring 8 and Photon Factory (Japan) synchrotron facilities. We thank Dr Torsten Juelich for linguistic assistance during the preparation of this manuscript. This work was supported by the Ministry of Science and Technology (China; 2014CB910104 to X.C.Z.), the Chinese Academy of Sciences (XDB080203 to X.C.Z. and F.S.) and National Natural Science Foundation of China (31470745 to X.C.Z.).

## Author contributions

F.S., Y.Z., and X.C.Z. supervised the project. Y.Xu and Y.Z. performed the crystallization experiments. G.L. and X.W. performed functional experiments. K.Z., Y.Xiong, Y.Xu, G.L. and H.L. performed the structural analyses. L.C. and J.L. performed the MD simulations. All authors discussed the experiments and contributed to the manuscript preparation. F.S. and X.C.Z. wrote the manuscript.

## Additional information

**Competing interests:** The authors declare no competing financial interests.

