## [Peer Review File · Nature Communications]

Reviewers' comments:

Reviewer #1 (Remarks to the Author):

The manuscript by Lu et al., reports the crystal structure of the essential Lnt-protein from *E. coli*, a highly conserved,

enzyme which is integral to the inner membrane and highly conserved in most, if not all, Gram-negative bacteria. The enzyme catalyzes the last enzymatic step during the N-terminal fatty-acylation of bacterial lipoproteins. While the biochemical properties of the enzyme are well studied, a better understanding of its unique enzymatic mechanism is lacking experimental structural data. This important, but very challenging task was successfully tackled by Lu et al.

In the present manuscript the authors report the crystal structure of this enzyme at a 2.6Å resolution. As predicted earlier, Lnt was revealed as a fusion of a periplasmic nitrilase domain that harbors the active site and disrupts the 8-TM helix domain that anchors the enzyme to the inner membrane and provides an embedded cavity for substrate entry and product release. Special attention is given to the flexible lid-loop of the nitrilase domain that can be regarded as a gate-keeper to regulate entry and exit. The authors point out that these structural studies are essential for a potential development of drugs targeting Lnt.

The manuscript is overall very well written and structured. The structural data are for the most part convincing and for the most part support the authors' conclusions. However, the accompanying biochemical data are of lower quality and partially contradicts previous findings (see below). Also the display of the structural data and order of their appearance makes it difficult for any reader to follow the authors' argumentation. This results largely from an overload of some supplementary data which should rather be integrated to one or two main figures and in accordance with the associated paragraphs of the results section. Overall, the authors primarily focus on the interactions with lipid substrate and fail to discuss if the structure of Lnt is compatible with the previously proposed ping-pong kinetics of the enzyme. These and other issues that deserve the authors' attention and need major improvement are listed below:

Results:

1. The authors report in I.110-114 that the purified Lnt protein was catalytically active in vitro and that first step and full reactions were analyzed. These data are presented in Fig. 1c+d. Do the authors refer to acyl-enzyme intermediate formation as "1st step"? I failed to find any data on the first reaction. The indicated reactions include all 3 reaction partners (PL, FSL-1 and enzyme). Please explain.

2. I.124: The authors have found Lnt to exist as monomer in solution. The monomer-weight by size-exclusion with detergents- which is very approximative - is given as 100 kDa in Figure S2. Dimer-formation seems unlikely, but can the authors really exclude it? The Fig. seems not convincing without control proteins.

3. I.123-141: One main finding is that the cavities of the TM and Nit domain are connected. The authors argue that this occurs through a highly conserved hydrogen bond between Y153 and R432. I failed to find any data on this and the role of these residues is not further addressed. Please explain.

4. The first two main paragraphs concerning the domain connection and the structural properties of the Nit domain are essentially only supported by Fig. 2. Instead the authors should integrate Suppl. Fig. 1c to the main results.

5. I.162-167: The authors should display structural data of the cavity to support their

argumentation that the conserved residues form the access site for both substrates.

6. Furthermore, a list of the 51 Lnt proteins that were used to identify conserved residues should be provided as well as the description of the alignment, instead of just referring to a website

7. The paragraph on the identification of essential residues in the central cavity dilutes the central information and

comprises a mixture of new information and confirmatory work, repeating previous assays on alkylation and in vivo complementation. This can be entirely moved to the supplements. At least 5 of these residues were identified previously by others and it is unclear why the authors have duplicated this work. Instead the authors should focus more on the structural impact of these residues and the ones they have newly identified, potentially by extending Fig. S5c to a higher resolution image. At present, this is provided as a supplement without any citation in the text.

8. The results section on the biochemical activity and stability of Lnt (I.224-232) is inadequate and seems to be without a logic connection to the structural data provided. In contrast to a previous kinetic study, the authors seem to have used one single end point measurement to conclude on the PL-headgroup specificity. Their finding that PG act as the most efficient FA donor contradicts previous kinetic data on the enzyme. It is also unclear, why the authors have studied the stability with different lipids instead of studying the impact of single residues that were identified as essential in the cavity. This section should be extended or entirely removed from a revised manuscript.

9. The interaction of the lid-loop with the lipid bilayer (I.233-271) is of high interest to understand the enzyme and the authors have approached this by a molecular dynamics simulation and conclude that the lid-loop makes intermediate contact to the lipid bilyer. Unfortunately the authors decided to put some of the most essential information into the supplements. Instead the supplements should rather include a video animation of the entire simulations which will make it easy for the reader to follow their argumentation. Especially as the authors report that Sim1 resulted in the entry of a lipid molecule to the cavity. This should be clearly visible in Fig. 4a or b.

10. The authors discuss mainly the flexibility of the lid-loop region, but it would be even more interesting whether their structural data is in accordance with the previously proposed ping-pong mechanism of the enzyme. How does the structure support the second step of the enzymatic mechanism, the transfer to the apolipoprotein? This has remained largely unaddressed.

11. The schematic diagram of Fig.5 is of poor quality and comprises no new information on the function of Lnt. The authors should at least include the dynamics of the lid loop and some basic structural characteristics of the enzyme.

12. Figure S5 shows an in-vivo screening for functionally important Lnt residues. Although the M&M indicate that expression depends on arabinose, which seems logic, legend and figure labels show glucose. Expression should be silent in glucose. Please explain.

Minor:

1. Fig. S1a: What was the selection of Lnts based on?
2. I. 96: citation of Buddelmeijer and Young is missing
3. I.272: Discussion
4. I.344+345: E. coli was not "transfected" but "transformed"
5. I.398-410: Lnt assay was carried out without detergent?
6. I.422-435: Lnt was shown to exist as a stable acyl-enzyme intermediate. Did the authors remove the acyl chain from the

- enzyme before adding the thiol-specific fluorophore? Was this assay done without detergent?
7. Fig.1: "Lgt" in the title of the figure legend?
 8. Fig. 1b: The labling with "vector" and "Lnt" is confusing? Do the authors refer to "wildtype" vs. Lnt under arabinose control? Why is this shown? The authors have described that the IPTG-inducible lac-promoter was used for overexpression of the enzyme.
 9. Fig.2: The rotation arrow is pointing in the wrong direction
 10. Fig. 4: A better color coding than blue and cyan would be useful here.

Reviewer #2 (Remarks to the Author):

MD simulations were performed by the authors to assess the functional role of the lid loop, in particular it's interaction with the lipid bilayer.

Major points:

-) figures in the manuscript are rather poor, it is quite difficult to capture the details of this novel x-ray structure with just 1 Figure in the main part of the manuscript! I strongly recommend including more details of the various structural elements of the protein. E.g. a detailed view of the Nit domain would be very helpful.
-) the authors state that they performed 3 independent MD simulations with different initial placements of the protein, however a detailed explanation of the rationale/methods of the different starting positions (and supplemental figures) are missing. It is questionable, if 3 presumably random placements can address the question posed by the authors. There are good methods available, concerning the placements of membrane proteins. For example coarse-grained simulations have proven very useful in this respect, and can be performed with very reasonable computational effort. (see Structure. 2008 Apr;16(4):621-30.)
-) Further it is not clear why the simulation time of Sim3 was stopped after 120ns?
-) page 9, line 244; The sentence "The protein maintained a tilted orientation relative to the membrane plane, with the cytosolic (periplasmic) portion of the curved TM4 being located about 45° (70°) off the direction of the membrane normal" is confusing. What does the value in brackets refer to?
-) "the results of the MD confirmed the possibility that Nit domain is partially embedded into the lipid": this statement is not sufficiently supported by the simulations, which besides being insufficiently detailed explained, are highly doubtful to give converged/robust answers to the question about the protein placement, due to the short length and number of repeats (3 x 120 – 150 ns).
-) Further the authors describe several interactions that stabilize protein – lipid interactions, but again no details are given. Monitoring the interactions over simulation time and showing this as supplemental plots would be very helpful.

Reviewer #3 (Remarks to the Author):

The manuscript by Lu et al describes the crystal structure of E coli apolipoprotein N-acyl transferase refined to 2.6Å resolution. This structure provides important insights into the enzymatic process of lipid modification of proteins. The structure is well determined and the manuscript is well written in general with a good overview of the structure and a detailed analysis of the residues implicated for catalysis. In my opinion this work is appropriate for publication to Nature Communications. I do have a list of correction and comments that the authors should consider in preparing a revised version of their manuscript.

One aspect of the structural analysis that is missing is a discussion of the thermal parameters of the protein. The MD simulation suggests that the most flexible region of the structure is the lid loop regions. Does this correlate with the temperature factors within the experimentally determined crystal structure?

In Figure 1a the rectangular box should be changed to R' to denote the different lipid donor molecules. The variations of R' should also be included in the figure. The ellipse in the figure denotes a diacylated apolipoprotein substrate and, as drawn, it is not adequately differentiated from the rectangular box.

On page 8 – mal-PEG should be properly defined.

Page 5 and Figure S2: the authors state that the SEC run in Figure S2 allow them to conclude that the protein is a monomer in solution. In order to really make these conclusions a SEC-MALS chromatography analysis would need to be performed. The SEC chromatogram does not allow for an estimate of the detergent micelle size while SEC-MALS does enable such an analysis to be made.

The SEC chromatogram only allows a conclusions to be made of the detergent/protein complex size, not whether the protein is a monomer or dimer.

Figure 2 needs to be more fully labelled. There are secondary structures being referred to in the text without a figure that displays where they lie in the overall structure. I acknowledge that this may be difficult to do for the nitilase domain however it may be work showing this domain separately as has also been done for the TM domain. In this case it will be easier to label all of the elements that are also being discussed in the text.

The authors are referring to a dome shaped cavity on page 6. In this context, and given the importance of this cavity, the manuscript would benefit from a better representation of this part of the structure. I would suggest making a cavity figure using the program CAVER.

In Figure S3a – define in the legend what the red starred residues refer to and what the residues with green circles above them refer to.

In Figure 3 the intensities of the bands differ significantly between panel a and panel b. This is particularly evident for the Lnt band in panel b. Can this be explained more fully. Also in panel b it appears as though multiple gels have been used to construct this panel. This experiment should be repeated so that all the mutants are analysed together on the same gel to ensure identical conditions.

The thermofluor assay in Figure S8 shows variable stability of the enzyme with different lipids. This is confusing as I would expect that the lipids would turnover by the enzyme. Thus in the timeframe in which the thermal shift assay is performed the enzyme would turnover the substrates. Was the assay carried out on the WT enzyme or an inactive mutant that is still able to bind substrates?

The MD simulation study shows that a lipid molecule is able to enter the cavity. Which of the two lipids making up the membrane in the simulation entered the cavity? Have the authors attempted to do any simulation using mutants that deter substrate binding, based on their activity assays, to show that, in these cases, the lipid does not enter the cavity?

The model proposed in Figure 5 suggests that a channel must exist for the N terminal diacylated lipoprotein to enter the active site. This channel must remain open to some extent to allow the chain to remain bound for catalysis. Is there any indication as to how the protein chain enters the active site. For example is a channel apparent in the structure? Also the N terminal amine group (the authors continually state this is an amide – but it should be an amine) would be expected to be triply protonated and thus positively charged. In order for the amine to react with the attack the reactive intermediate, a proton would have to be removed so that the substrate can act as a nucleophile. Figure 5 does not give any indication as to how this occurs. Nor is there any discussion of this in the manuscript. Given this lack of detail, I think Figure 5 is not informative enough to be included in the manuscript.

Minor Corrections

Line 50: "... as pre-prolipoproteins in cytosol." Should be "... as pre-prolipoproteins in the cytosol."

Line 65 – I think the authors mean to refer to the alpha-amino group, not the alpha amide.

Line 68: "Lnt belongs to the ninth of 13 branches of nitrilase (Nit) superfamily ..." should be "Lnt belongs to the ninth of 13 branches of the nitrilase (Nit) superfamily..."

Line 86 "...namely acyl donor and acceptor..." should be "...namely an acyl donor and an acceptor..."

Line 94/95: "activate nucleophileand to activate amide group...." Should be "activate the nucleophileand to activate the amine group...."

Line 99: "...using x-ray crystallography" should be "...using X-ray crystallography"

Figure S4 – remove the ellipse from the first panel. It does not add to that panel and is redundant with the second last panel in the figure which is where the interfacial opening is being shown.

Line 227 – define FSL-1.

Line 326 "... uses its N-terminal amide group....." should be "... uses its N-terminal amine group....."

Reviewers' comments:

Reviewer #1 (Remarks to the Author):

The manuscript by Lu et al., reports the crystal structure of the essential Lnt-protein from *E. coli*, a highly conserved, enzyme which is integral to the inner membrane and highly conserved in most, if not all, Gram-negative bacteria. The enzyme catalyzes the last enzymatic step during the N-terminal fatty-acylation of bacterial lipoproteins. While the biochemical properties of the enzyme are well studied, a better understanding of its unique enzymatic mechanism is lacking experimental structural data. This important, but very challenging task was successfully tackled by Lu et al.

In the present manuscript the authors report the crystal structure of this enzyme at a 2.6Å resolution. As predicted earlier, Lnt was revealed as a fusion of a periplasmic nitrilase domain that harbors the active site and disrupts the 8-TM helix domain that anchors the enzyme to the inner membrane and provides an embedded cavity for substrate entry and product release. Special attention is given to the flexible lid-loop of the nitrilase domain that can be regarded as a gate-keeper to regulate entry and exit. The authors point out that these structural studies are essential for a potential development of drugs targeting Lnt.

The manuscript is overall very well written and structured. The structural data are for the most part convincing and for the most part support the authors' conclusions. However, the accompanying biochemical data are of lower quality and partially contradicts previous findings (see below). Also the display of the structural data and order of their appearance makes it difficult for any reader to follow the authors' argumentation. This results largely from an overload of some supplementary data which should rather be integrated to one or two main figures and in accordance with the associated paragraphs of the results section. Overall, the authors primarily focus on the interactions with lipid substrate and fail to discuss if the structure of Lnt is compatible with the previously proposed ping-pong kinetics of the enzyme. These and other issues that deserve the authors' attention and need major improvement are listed below:

Results:

1. The authors report in l.110-114 that the purified Lnt protein was catalytically active in vitro and that first step and full reactions were analyzed. These data are presented in Fig. 1c+d. Do the authors refer to acyl-enzyme intermediate formation as "1st step"? I failed to find any data on the first reaction. The indicated reactions include all 3 reaction partners (PL, FSL-1 and enzyme). Please explain.

Thanks for the reviewer to point out this problem. Indeed, in Figure 1, we did not show the formation of the acyl-enzyme intermediate. We have made change in the revision.

Figure 3a shows that WT Lnt forms intermediate in the first step reaction, which inhibits the formation of Lnt-Mal-PEG covalent compound.

2. I.124: The author have found Lnt to exist as monomer in solution. The monomer-weight by size-exclusion with detergents- which is very approximative - is given as 100 kDa in Figure S2. Dimer-formation seems unlikely, but can the authors really exclude it? The Fig. seems not convincing without control proteins.

We add BSA as a control, which shows two peaks at 66 and 132 kDa. In addition, we also add a SEC-MALS assay which also showed that monomer is the dominant specie of Lnt. Hopefully, these may resolve the reviewer's concern.

3.I.123-141: One main finding is that the cavities of the TM and Nit domain are connected. The authors argue that this occurs through a highly conserved hydrogen bond between Y153 and R432. I failed to find any data on this and the role of these residues is not further adressed. Please explain.

We add Figure 3b to show the local structure of these residues. Moreover, we tested mutations at these two positions (one of them is added in the revision), showing either abolished or reduced *in vivo* activity.

4. The first two main paragraphs concerning the domain connection and the structural properties of the Nit domain are essentially only supported by Fig. 2. Instead the authors should integrate Suppl. Fig. 1c to the main results.

As the reviewer suggested, we move the supplementary figure into the main text.

5.I.162-167: The authors should display structural data of the cavity to support their argumentation that the conserved residues form the access site for both substrates.

Figure 2 is added to support this point. Also see Figure S3b.

6. Furthermore, a list of the 51 Lnt proteins that were used to identify conserved residues should be provided as well as the description of the alignment, instead of just referring to a website

We rechecked the alignment of the PFAM and found that the web logo plot was actually based on 22 Lnt sequences. The Uniprot IDs of these proteins are given in the figure legend of revision.

7. The paragraph on the identification of essential residues in the central cavity dilutes the central information and comprises a mixture of new information and confirmatory work, repeating previous assays on alkylation and *in vivo* complementation. This can be

entirely moved to the supplements. At least 5 of these residues were identified previously by others and it is unclear why the authors have duplicated this work.

These five mutants were included here for the purpose of showing that our assay was corrected setup.

Instead the authors should focus more on the structural impact of these residues and the ones they have newly identified, potentially by extending Fig. S5c to a higher resolution image. At present, this is provided as a supplement without any citation in the text.

Figure 2 is added to the main text to replace Fig. S5c.

8. The results section on the biochemical activity and stability of Lnt (I.224-232) is inadequate and seems to be without a logic connection to the structural data provided. In contrast to a previous kinetic study, the authors seem to have used one single end point measurement to conclude on the PL-headgroup specificity. Their finding that PG act as the most efficient FA donor contradicts previous kinetic data on the enzyme. It is also unclear, why the authors have studied the stability with different lipids instead of studying the impact of single residues that were identified as essential in the cavity. This section should be extended or entirely removed from a revised manuscript.

We agree with the reviewer in that more kinetic data might be needed to conclude whether PE or PG is a better substrate. Since a complete kinetic study would be beyond the scope of the manuscript, we choose to soften all statements on PG.

9. The interaction of the lid-loop with the lipid bilayer (I.233-271) is of high interest to understand the enzyme and the authors have approached this by a molecular dynamics simulation and conclude that the lid-loop makes intermediate contact to the lipid bilayer. Unfortunately the authors decided to put some of the most essential information into the supplements. Instead the supplements should rather include a video animation of the entire simulations which will make it easy for the reader to follow their argumentation. Especially as the authors report that Sim1 resulted in the entry of a lipid molecule to the cavity. This should be clearly visible in Fig. 4a or b.

A video has been added as supplementary material.

10. The authors discuss mainly the flexibility of the lid-loop region, but it would be even more interesting whether their structural data is in accordance with the previously proposed ping-pong mechanism of the enzyme. How does the structure support the second step of the enzymatic mechanism, the transfer to the apolipoprotein? This has remained largely unaddressed.

We tend to believe that the ping-pong mechanism is correct. However, the current structure does not provide a direct support to the mechanism. We plan to work on structures of enzyme-substrate complexes. Hopefully, we will be able to address these

questions in the future.

11. The schematic diagram of Fig.5 is of poor quality and comprises no new information on the function of Lnt. The authors should at least include the dynamics of the lid loop and some basic structural characteristics of the enzyme.

At this point, there is no data to support an active role of the lip loop in regulating the reaction dynamics. Its role is more likely to protect the active site from the solvent access. We move the schematic diagram to the supplementary material.

12. Figure S5 shows an in-vivo screening for functionally important Lnt residues. Although the M&M indicate that expression depends on arabinose, which seems logic, legend and figure labels show glucose. Expression should be silent in glucose. Please explain.

In the revision, we add explanations in the legend for Fig. 1b to explain the in-vivo screening, and in the legend of Fig. S5 we pointed out the connection to Fig 1b.

In the presence of glucose, the rescuing plasmid (containing WT Lnt gene) is inhibited, and the Lnt activity comes from another plasmid (containing the gene of an Lnt variant).

Minor:

1. Fig. S1a: What was the selection of Lnts based on?

More information has been included in the figure legend.

2. l. 96: citation of Buddelmeijer and Young is missing

Done

3. l.272: Discussion

Changed

4. l.344+345: E. coli was not "transfected" but "transformed"

Changed

5. l.398-410: Lnt assay was carried out without detergent?

There was 0.2% DM in the reaction mixture. Change has been made in the revision.

6. l.422-435: Lnt was shown to exist as a stable acyl-enzyme intermediate. Did the authors remove the acyl chain from the enzyme before adding the thiol-specific fluorophore? Was this assay done without detergent?

In this particular experiment, we did not remove the acyl chain from the purified enzyme. To our knowledge, currently there is no good procedure to remove such acyl chains without disrupting the Lnt structure.

Regarding the detergent, see point 5.

7. Fig.1: "Lgt" in the title of the figure legend?

Changed

8. Fig. 1b: The labling with "vector" and "Lnt" is confusing? Do the authors refer to "wildtype" vs. Lnt under arabinose control? Why is this shown?

'Vector' stands for a plasmid without the Lnt gene. It is included as a negative control showing that in the presence of glucose (inhibitor of the arabinose-inducible plasmid) WT Lnt rescues the Δlnt cell strain. (see ref. (Robichon et al., 2005))

Change is made in the revision.

The authors have described that the IPTG-inducible lac-promoter was used for overexpression of the enzyme.

In this particular assay, Lnt protein was expressed from the pEXS-DH plasmid, which is a constitutive expression vector.

In addition, a different IPTG-inducible plasmid was used to overexpress Lnt for the crystallization experiment.

9. Fig.2: The rotation arrow is pointing in the wrong direction

We rechecked the arrow, which seems correct. The Fig. 2b shows the TM domain, with the Nit domain being removed.

10. Fig. 4: A better color coding than blue and cyan would be useful here.

Changed

Reviewer #2 (Remarks to the Author):

MD simulations were performed by the authors to assess the functional role of the lid loop, in particular it's interaction with the lipid bilayer.

Major points:

-) figures in the manuscript are rather poor, it is quite difficult to capture the details of this novel x-ray structure with just 1 Figure in the main part of the manuscript! I strongly recommend including more details of the various structural elements of the protein. E.g. a detailed view of the Nit domain would be very helpful.

Figure 2 is added.

-) the authors state that they performed 3 independent MD simulations with different initial placements of the protein, however a detailed explanation of the rationale/methods of the different starting positions (and supplemental figures) are missing. It is questionable, if 3 presumably random placements can address the question posed by the authors. There are good methods available, concerning the placements of membrane proteins. For example coarse-grained simulations have proven very useful in this respect, and can be performed with very reasonable computational effort. (see Structure. 2008 Apr;16(4):621-30.)

We agree with the reviewer in that placing protein into membrane is an important issue in MD simulations. In this manuscript, the initial membrane was generated with the CHARMM-GUI membrane builder, and was pre-equilibrated for more than 100 ns. At first, Lnt protein was placed into the membrane with a strategy suggested in NAMD's membrane protein tutorial (<http://www.ks.uiuc.edu/Training/Tutorials/science/membrane/mem-tutorial.pdf>), which is not completely RANDOM. This results in the initial configuration of Sim2. The same procedure of protein insertion has been used for several membrane simulations. Sim1 and Sim3 are generated manually by rotate the protein 5° toward two opposite directions to make the LID loop contact (Sim1) or away (Sim3) to the membrane. One aim of our simulation was to check the possible interactions between the Lid loop and the membrane, and thus we adopted the membrane insertion method described above.

Nevertheless, this membrane insertion procedure is sort of artificial. In order to reduce possible artifacts, for all three initial configurations, we did careful minimization and equilibration. In particular, 20 ns simulation was carried out with the C α atom of the protein fixed to let the lipid molecules adjust their position around. We believe that this process removed the possible artifacts being induced by initial protein placing.

In this revision, we have added more description to clarify the procedural, as the reviewer suggested. We also agree that the coarse-grained membrane-insertion procedure is useful and helpful, and we plan to use such a technique in further.

-) Further it is not clear why the simulation time of Sim3 was stopped after 120ns?

The simulation of Sim3 only reach 120 ns when we submits the manuscript, thus we only report 120 ns results for it. All three simulations are now reported for 150 ns long. We have updated the results in the revised manuscript.

-) page 9, line 244; The sentence “The protein maintained a tilted orientation relative to the membrane plane, with the cytosolic (periplasmic) portion of the curved TM4 being located about 45° (70°) off the direction of the membrane normal” is confusing. What does the value in brackets refer to?

Changes are made to clarify this.

-) “the results of the MD confirmed the possibility that Nit domain is partially embedded into the lipid”: this statement is not sufficiently supported by the simulations, which besides being insufficiently detailed explained, are highly doubtful to give converged/robust answers to the question about the protein placement, due to the short length and number of repeats (3 x 120 – 150 ns).

We agree with the reviewer. Thus, we have soften our statement, replacing “confirmed the possibility” with “ are in agreement with the suggested possibility”.

-) Further the authors describe several interactions that stabilize protein – lipid interactions, but again no details are given. Monitoring the interactions over simulation time and showing this as supplemental plots would be very helpful.

Two panels are added in Fig. 5 showing the lid-loop conformational dynamics at different time points. In addition a movie is added as supplementary material to show the dynamics of the Lnt protein in the lipid bilayer.

Reviewer #3 (Remarks to the Author):

The manuscript by Lu et al describes the crystal structure of E coli apolipoprotein N-acyl transferase refined to 2.6Å resolution. This structure provides important insights into the enzymatic process of lipid modification of proteins. The structure is well determined and the manuscript is well written in general with a good overview of the structure and a detailed analysis of the residues implicated for catalysis. In my opinion this work is appropriate for publication to Nature Communications. I do have a list of correction and comments that the authors should consider in preparing a revised version of their manuscript.

One aspect of the structural analysis that is missing is a discussion of the thermal parameters of the protein. The MD simulation suggests that the most flexible region of the structure is the lid loop regions. Does this correlate with the temperature factors within the experimentally determined crystal structure?

Indeed, the lid loop region is of high B factor. Part of the loop was not built because of lack of electron density.

In Figure 1a the rectangular box should be changed to R' to denote the different lipid donor molecules. The variations of R' should also be included in the figure. The ellipse in the figure denotes a diacylated apolipoprotein substrate and, as drawn, it is not adequately differentiated from the rectangular box.

Under Figure 1a, we add legends to differentiate the rectangular box and ellipse as a lipid substrate and apolipoprotein, respectively.

On page 8 – mal-PEG should be properly defined.

Done

Page 5 and Figure S2: the authors state that the SEC run in Figure S2 allow them to conclude that the protein is a monomer in solution. In order to really make these conclusions a SEC-MALS chromatography analysis would need to be performed. The SEC chromatogram does not allow for an estimate of the detergent micelle size while SEC-MALS does enable such an analysis to be made. The SEC chromatogram only allows a conclusions to be made of the detergent/protein complex size, not whether the protein is a monomer or dimer.

We add a control sample to this SEC experiment, which confirms that Lnt/detergent complex is of a size of ~100 kDa. Furthermore, as suggested by the reviewer, a SEC-MALS assay is added which also indicates that monomer is the dominant specie of Lnt.

Figure 2 needs to be more fully labelled. There are secondary structures being referred to in the text without a figure that displays were they lie in the overall structure. I

acknowledge that this may be difficult to do for the nitilase domain however it may be work showing this domain separately as has also been done for the TM domain. In this case it will be easier to label all of the elements that are also being discussed in the text.

We move a figure on Nit domain to the main text, which contains some labels.

The authors are referring to a dome shaped cavity on page 6. In this context, and given the importance of this cavity, the manuscript would benefit from a better representation of this part of the structure. I would suggest making a cavity figure using the program CAVER.

A figure (Figure 2e in the revision) has been added to show the shape of the substrate-binding cavity.

In Figure S3a – define in the legend what the red starred residues refer to and what the residues with green circles above them refer to.

Done.

In Figure 3 the intensities of the bands differ significantly between panel a and panel b. This is particularly evident for the Lnt band in panel b. Can this be explained more fully. Also in panel b it appears as though multiple gels have been used to construct this panel. This experiment should be repeated so that all the mutants are analysed together on the same gel to ensure identical conditions.

In the original manuscript, the two panels were obtained with different techniques (scanning vs film exposure). We remade this figure (now Figure 4) using a scanner.

The thermofluor assay in Figure S8 shows variable stability of the enzyme with different lipids. This is confusing as I would expect that the lipids would turnover by the enzyme. Thus in the timeframe in which the thermal shift assay is performed the enzyme would turnover the substrates. Was the assay carried out on the WT enzyme or an inactive mutant that is still able to bind substrates?

The thermofluor assay was carried out with WT Lnt. However, since we did not add the second substrate apolipoprotein, the reaction could not be fully completed.

The MD simulation study shows that a lipid molecule is able to enter the cavity. Which of the two lipids making up the membrane in the simulation entered the cavity? Have the authors attempted to do any simulation using mutants that deter substrate binding, based on their activity assays, to show that, in these cases, the lipid does not enter the cavity?

In Sim1, a DPPE molecule entered the cavity, which suggests the possibility that the cavity can accommodate lipid molecules. Nevertheless, it is not observed in Sim2 and

Sim3. Due to the limited time-scale of MD simulation, if an event is not observed in MD doesn't necessarily mean it could not happen. Thus, we did not perform any mutant simulations. In our opinion, we would not be able to make a conclusion, based on MD simulations alone, that lipid molecules did not enter the cavity even if no lipid entering were observed in all our simulations with mutants.

The model proposed in Figure 5 suggests that a channel must exist for the N terminal diacylated lipoprotein to enter the active site. This channel must remain open to some extent to allow the chain to remain bound for catalysis. Is there any indication as to how the protein chain enters the active site. For example is a channel apparent in the structure?

Indeed, the substrate-binding cavity (see Figure 2e) seems to remain open to the lipid bilayer.

Also the N terminal amine group (the authors continually state this is an amide – but it should be an amine) would be expected to be triply protonated and thus positively charged. In order for the amine to react with the attack the reactive intermediate, a proton would have to be removed so that the substrate can act as a nucleophile. Figure 5 does not give any indication as to how this occurs. Nor is there any discussion of this in the manuscript. Given this lack of detail, I think Figure 5 is not informative enough to be included in the manuscript.

The reviewer raises an interesting question on the catalytic mechanism. Since the current structure contains neither of two substrates, we choose not to discuss such detailed mechanism.

(Old) Figure 5 has been removed from the main text.

Minor Corrections

Line 50: "... as pre-prolipoproteins in cytosol." Should be "... as pre-prolipoproteins in the cytosol."

Changed

Line 65 – I think the authors mean to refer to the alpha-amino group, not the alpha amide.

Changed

Line 68: "Lnt belongs to the ninth of 13 branches of nitrilase (Nit) superfamily ..." should be "Lnt belongs to the ninth of 13 branches of the nitrilase (Nit) superfamily..."

Changed

Line 86 "...namely acyl donor and acceptor..." should be "...namely an acyl donor and an acceptor..."

Changed

Line 94/95: "activate nucleophileand to activate amide group...." Should be "activate the nucleophileand to activate the amine group...."

Changed

Line 99: "...using x-ray crystallography" should be "...using X-ray crystallography"

Changed

Figure S4 – remove the ellipse from the first panel. It does not add to that panel and is redundant with the second last panel in the figure which is where the interfacial opening is being shown.

Done

Line 227 – define FSL-1.

Done

Line 326 "... uses its N-terminal amide group....." should be "... uses its N-terminal amine group....."

Changed

REVIEWERS' COMMENTS:

Reviewer #1 (Remarks to the Author):

The authors greatly improved the previously submitted manuscript and responded to all my concerns and questions. As this challenging work is technically sound and its outcome of very high interest to a wide readership I support its publication.

Reviewer #2 (Remarks to the Author):

The authors have appropriately addressed all my concerns.